# Integrated Biological Control Using a Mixture of Two Entomopathogenic Bacteria, *Bacillus thuringiensis* and *Xenorhabdus hominickii*, against *Spodoptera exigua* and Other Congeners

**DOI:** 10.3390/insects13100860

**Published:** 2022-09-21

**Authors:** Md Tafim Hossain Hrithik, Youngjin Park, Hyemi Park, Yonggyun Kim

**Affiliations:** 1Department of Plant Medicals, College of Life Sciences, Andong National University, Andong 36729, Korea; 2Animal and Plant Quarantine Agency, 167, Yongjeon-ro, Gimcheon 39660, Korea

**Keywords:** *Bacillus thuringiensis aizawai*, *Spodoptera exigua*, *Xenorhabdus hominickii*, secondary metabolite, biological control

## Abstract

**Simple Summary:**

An entomopathogen, *Bacillus thuringiensis* (Bt), has been used to control insect pests. On the other hand, insect immune responses defend the bacterial pathogenicity. An idea to enhance the Bt virulence was to inhibit insect immune defense using immunosuppressant. Secondary metabolites from another entomopathogen, *Xenorhabdus hominickii*, are known to inhibit the insect immune responses, but they alone give little virulence against insects. The addition of the bacterial culture broth containing the secondary metabolites to Bt spores significantly enhanced the Bt virulence in laboratory and field trials. This study demonstrates an integrated biological control by ideally combining two entomopathogenic bacteria.

**Abstract:**

Insect immunity defends against the virulence of various entomopathogens, including *Bacillus thuringiensis* (Bt). This study tested a hypothesis that any suppression of immune responses enhances Bt virulence. In a previous study, the entomopathogenic bacterium, *Xenorhabdus hominickii* (Xh), was shown to produce secondary metabolites to suppress insect immune responses. Indeed, the addition of Xh culture broth (XhE) significantly enhanced the insecticidal activity of Bt against *S. exigua*. To analyze the virulence enhanced by the addition of Xh metabolites, four bacterial secondary metabolites were individually added to the Bt treatment. Each metabolite significantly enhanced the Bt insecticidal activity, along with significant suppression of the induced immune responses. A bacterial mixture was prepared by adding freeze-dried XhE to Bt spores, and the optimal mixture ratio to kill the insects was determined. The formulated bacterial mixture was applied to *S. exigua* larvae infesting Welsh onions in a greenhouse and showed enhanced control efficacy compared to Bt alone. The bacterial mixture was also effective in controlling other Spodopteran species such as *S. litura* and *S. frugiperda* but not other insect genera or orders. This suggests that Bt+XhE can effectively control *Spodoptera*-associated pests by suppressing the immune defenses.

## 1. Introduction

Two bacterial genera, *Xenorhabdus* and *Photorhabdus*, have a species-specific mutualistic symbiosis with their host entomopathogenic nematodes, *Steinernema* and *Heterorhabditis*, respectively [1]. These bacteria encode a biosynthetic gene cluster in their genomes and produce secondary metabolites with virulence against parasitic hosts [2]. The bacteria reside in the intestine of the nematodes at their infective juvenile (IJ) stage. When IJ nematodes enter the target insect hemocoel through natural openings such as the mouth, anus, and spiracles, they release the specific bacteria from the intestine [3]. The released bacteria in the hemocoel grow and sequentially release secondary metabolites to inhibit insect immune responses [4]. Both bacterial genera commonly inhibit eicosanoid biosynthesis of the target insects [5].

Eicosanoids are a group of oxygenated C20 polyunsaturated fatty acids that mediate various physiological processes, including immunity in insects [6]. Insect immunity is characterized by innately programmed responses consisting of cellular and humoral components [7]. Upon immune challenge, pattern recognition receptors recognize specific microbes, and the recognition signal is subsequently propagated to nearby or local immunity-associated tissues such as hemocytes or fat bodies via autocrine or paracrine signaling using immune mediators such as biogenic monoamines, cytokines, and eicosanoids [8]. Cross-talk among immune mediators leads to an up-regulation of eicosanoid biosynthesis to produce different types of eicosanoids such as prostaglandins (PGs), leukotrienes, and epoxyeicosatrienoic acids to mediate various immune responses [9,10]. Although most PGs and other eicosanoids stimulate different immune responses upon immune challenge, a specific eicosanoid, PGI_2_, which is produced in the late phase of an immune response, suppresses unnecessary and excessive immune responses [11]. Thus, the specific inhibition of eicosanoid biosynthesis by *Xenorhabdus* and *Photorhabdus* interrupts the complex interplay between the immune defenses of the target insects, which would be favorable for survival and the development of the bacterial-nematode complex [12].

As an alternative to chemical insecticides, which cause serious environmental adverse effects, biological control agents have been widely adopted to control pest insects. *Bacillus thuringiensis* (Bt) is one of the successful biological control agents [13]. Bt produces crystal (Cry) toxin, which binds to receptor(s) on the epithelial membrane of the insect midgut to form pores, and the microbes in the gut lumen enter through the pores to proliferate in the hemocoel, leading to septicemia [14]. However, its wide practical application has limitations because of its relatively slow speed-to-kill compared to chemical insecticides and narrow control spectrum due to its pathogenic specificity. Furthermore, several insect pests have developed Bt resistance, which hinders the control efficacy by increasing proteolytic activity in the gut lumen and altering Bt-binding site(s) or insect immune responses [15,16]. The concept of integrated biological control (IBC) was devised to effectively control insect pests by mixing two or more biological control agents with different pathogenic molecular targets [17]. For example, the virulence of the entomopathogenic Bt bacterium was highly enhanced against the lepidopteran pest *Spodoptera exigua* by mixing Bt with *X. nematophila* [18]. *Xenorhandus nematophila* is a mutualistic bacterium of the entomopathogenic nematode *Steinernema carpocapsae* that inhibits the immune responses of target insects in their hemocoel [19]. In mixed bacterial treatment, the pore-forming activity of Bt in the gut epithelium allows *X. nematophila* to enter the hemocoel of the target insects, in which the immunosuppressive conditions induced by *X. nematophila* enhance Bt virulence by suppressing the immune defenses of the target insects. The immunosuppressive activity of *X. nematophila* was applied to a baculoviral pathogen against *Plutella xylostella* [20] and the fungal pathogen *Beauveria bassiana* against *S. exigua* [21] as other examples of IBC strategies. This IBC approach has also been applied to non-lepidopteran insect pests such as the coleopteran *Phaedon brassicae* [22] and mosquitoes *Aedes albopictus* and *Culex pipiens pallens* [23]. Due to the high susceptibility of *Xenorhabdus* and *Photorhabdus* to ultraviolet (UV) irradiation, desiccation, or occasional high temperatures in the agricultural environment, bacterial culture broth containing secondary metabolites that suppress insect immunity has been used to prepare Bt mixtures instead of the bacterial cells themselves [24]. Mastore et al. [25] demonstrated that the bacterial culture broth of *X. nematophila* significantly enhanced Bt virulence against the dipteran *Drosophila suzukii*.

Some insect species in the *Spodoptera* genus significantly damage crops. In Korea, *S. exigua* and *S. litura* infest various vegetables and ornamental crops [26]. However, it is difficult to control these insects due to their resistance to chemical insecticides [27,28]. Recently, *S. frugiperda* invasions were reported in western Korea, intimidating the crop yields of rice and corn [29]. A highly virulent strain of *X. hominickii* was found from a comparative toxicological analysis and used to replace a previous IBC agent to control *S. exigua* [30]. Thus, this study was devised to develop a highly efficient IBC agent against spodopteran insect pests without hazards to humans or the environment.

## 2. Materials and Methods

### 2.1. Insect Rearing

*Spodoptera exigua* larvae were collected from a Welsh onion (*Allium fistulosum* L.) field in Andong, Korea, and reared with an artificial diet [31]. The mealworm, *Tenebrio molitor*, was provided by Bio Utility (Andong, Korea). *P. xylostella* larvae were collected from cabbage (*Brassica rapa* L.) fields in Andong and reared with a diet of fresh cabbage. The legume pod borer, *Maruca vitrata*, was collected from an adzuki field in Suwon, Korea, and reared with an artificial diet [32]. The brassica leaf beetle, *P. brassicae*, was reared with a diet of cabbage. *S. frugiperda* larvae were obtained from Frontier Agriculture Sciences (Newark, DE, USA) and maintained in the Quarantine Pest Research Facility of the Animal and Plant Quarantine Agency [33]. *A. albopictus* mosquito larvae (ca. 200/pan) were donated from Prof. Yeon Ho Je (Seoul National University, Seoul, Korea) and fed 100 mg of ground fish food (Millepet Artemia, Ansan, Korea) with dried yeast (Samchon Chemical, Damyang, Korea) per 500 mL of water daily. Sugar solution (10%, *w*/*v*) was continuously provided to adult males and females. Females were provided with blood using mice until engorged. Freshly laid eggs were collected, washed with 0.1% bleach, and hatched in distilled water. All insects were reared at 25 ± 2 °C, relative humidity of 60 ± 5%, and 16:8 (L:D) photoperiod. In these rearing conditions, *S. exigua* underwent five larval molts (L1–L5). Adults were fed 10% sugar solution. Larvae at L3 stage were used for the bioassays, while L5 larvae were used for immunological assays.

### 2.2. Chemicals

Benzylideneacetone (BZA) and oxindole (OXI) were purchased from Sigma-Aldrich Korea (Seoul, Korea). GameXPeptide-A (GXP) was provided by Professor Helge Bode (Max Planck Institute, Marburg, Germany) and 3-ethoxy-4-methoxyphenol (EMP) was purchased from BOC Sciences (Shirly, NY, USA). These compounds were dissolved in acetone to make the stock solutions and further diluted with 5% acetone containing 0.01% Triton X-100 for the treatments.

### 2.3. Bacterial Culture

*Escherichia coli* Top10 was obtained from Invitrogen (Carlsbad, CA, USA) and cultured in Luria-Bertani (LB) medium (BD, Franklin Lakes, NJ, USA) overnight at 37 °C. *X. hominickii* ANU101 (Xh) isolated from *Steinernema monticolum* [34] was cultured in tryptic soy broth (TSB, Difco, Sparks, MD, USA) for 48 h at 28 °C in a shaking incubator at 160 rpm. The culture broth was then freeze-dried, in which 1 L culture broth yielded around 3 g of dried powder (called Xh extract (XhE)). The sources of the four Bt subspecies of *B. thuringiensis kurstaki*, *B. thuringiensis aizawai* (BtA), *B. thuringiensis israelensis* (BtI), and *B. thuringiensis tenebrionis* (BtT) were previously described [35]. The Bt subspecies were cultured in TSB under the same conditions as described above. After 48 h of culture, the cultured Bt suspension was post-treated at 4 °C for another 48 h for sporulation. After centrifugation at 8000 rpm, the cell pellets were harvested and freeze-dried, in which 1 L bacterial culture yielded around 0.65 g.

### 2.4. RNA Extraction, cDNA Preparation, and qPCR

The L5 larvae of *S. exigua* were used to collect different tissues. Total RNA was extracted from the fat body using Trizol reagent (Invitrogen, Carlsbad, CA, USA) according to the manufacturer’s instructions. Extracted RNA was used for synthesizing complementary DNA (cDNA) using an RT-premix (Intron Biotechnology, Seoul, Korea) containing oligo-dT primers. Synthesized cDNA was quantified using a spectrophotometer (NanoDrop, Thermo Fisher Scientific, Wilmington, DE, USA). The synthesized cDNA (80 ng per µL) was used as a template for quantitative PCR (qPCR) using gene-specific primers (Appendix A). qPCR was performed using SYBR Green Real-time PCR Master Mix (Toyobo, Osaka, Japan) according to the guidelines of Bustin et al. [36] on a real-time PCR system (Step One Plus Real-Time PCR System, Applied Biosystem, Singapore). The reaction mixture (20 μL) contained 10 μL of Power SYBR Green PCR Master Mix, 1 μL of cDNA template (80 ng), 1 μL each of forward and reverse primers (Appendix A), and 7 µL of deionized distilled water. The temperature program for qPCR began with a 95 °C heat treatment for 10 min followed by 40 cycles of denaturation at 94 °C for 30 s, annealing at 52 °C for 1 min, and extension at 72 °C for 30 s. The ribosomal gene, RL32, was used to validate cDNA integrity. Each biological treatment was replicated three times by individual tissue preparations. qPCR expression was calculated by the comparative CT method [37].

### 2.5. Nodulation Assay

Two-day-old L5 larvae were used for performing the nodule formation assay. Heat-killed E. coli were injected at a dose of 1.75 × 10^5^ cells/larva through the abdominal proleg using a 5-µL microsyringe (Hamilton, Reno, NV, USA). After 8 h of incubation at 25 °C, the formed nodules were counted by dissecting the larvae under a stereomicroscope (Stemi SV 11, Zeiss, Jena, Germany) at 50× magnification. Each treatment was replicated three times. Three larvae were used for each replication.

### 2.6. Antimicrobial Peptide (AMP) Gene Expression Analysis

To assess AMP gene expression, *S. exigua* L5 larvae were treated with heat-killed *E. coli* as described above. Eight hours post-treatment, the fat bodies were collected for RNA extraction and cDNA synthesis. The expression levels of 10 AMP genes (apolipophorin III, attacin 1, attacin 2, defensin, gallerimycin, gloverin, lysozyme, transferrin I, transferrin II, and cecropin) were assessed by RT-qPCR using gene-specific primers (Appendix A). The ribosomal gene, *RL32*, was used as a reference gene to normalize the target gene expression levels. Each treatment was independently replicated three times and each replication consisted of three individual larvae.

### 2.7. Bt Bioassays

The four different Bt strains were prepared at different concentrations of 0, 10^6^, 10^7^, 10^8^, and 10^9^ spores/mL. L3 instar larvae from the four different insect species (*S. exigua*, *P. xylostella*, *T. molitor*, and *A. albopictus*) were used for the Bt bioassays, in which the leaf-dipping method was used for all bioassays except for the mosquito assay. Briefly, a piece of cabbage leaf (2 cm^2^) was dipped into a Bt suspension for 5 min. After removing excess suspension, the treated cabbage was placed on filter paper in a Petri dish (90 × 15 mm), on which 10 randomly chosen larvae were released. After 24 h, the treated leaf was replaced with a fresh one. Data were collected daily for up to 3 days. Each concentration was replicated three times with 10 larvae per replication. For the mosquito bioassay, early L3 larvae were used for all bioassays in 1 mL of distilled water suspension containing different concentrations of BtI. For the dose–mortality test, BtI was prepared at 0, 10^6^, 10^7^, 10^8^, and 10^9^ spores/mL. Each bioassay was replicated three times. Each replication used 10 individual larvae per well of a 12-well Costar culture plate (Corning, Lowell, MA, USA). The plates were placed at 25 °C with a relative humidity of 70% in an environmental chamber. Larval mortalities were assessed 3 days after treatment.

### 2.8. Bioassay with BtA+Bacterial Metabolites

Different concentrations of bacterial metabolites were added to a fixed concentration (108 spores/mL) of BtA. The bacterial metabolites included XhE, and four bacterial metabolites (GXP, EMP, BZA, and OXI). XhE was tested at 1, 2, 3, 4, and 5 mg/mL. Metabolites were tested at 0.001, 0.01, 0.1, and 1 mg/mL. The bioassays were performed using the leaf-dipping method described above.

### 2.9. Field Assay to Estimate Control Efficacy

To assess the control efficacy of BtA+XhE mixtures in field conditions, an experiment was performed in a greenhouse in a randomized block design with three replicates. Welsh onions were grown in nine rows, in which the row-to-row distance was around 30 cm and the plant-to-plant distance was around 10 cm. Onions were infested with the young (L2–L3) larvae of *S. exigua*. Just before spraying the test insecticides, the total numbers of larvae per plant were 26~31. A mixture of BtA (1.25 mg (=10^8^ spores)/mL) and XhE (4 mg/mL) was prepared in a mass ratio. For the positive control, BtA alone was sprayed. For the negative control, 5% acetone in water was sprayed. Each treatment was sprayed in a volume of 70 mL in each row.

### 2.10. Control Spectrum of BtA+XhE Mixture

The leaf-dipping method was used to determine the control efficacies of the BtA+XhE mixture against different insects, including *S. exigua*, *S. litura*, *S. frugiperda*, *M. vitrata*, *P. xylostella*, *P. brassicae*, and *T. molitor*, but not *A. albopictus*. An artificial diet was used for the dipping assay against *S. litura*, *S. frugiperda*, and *M. vitrata*. For *P. xylostella*, *P. brassicae*, and *T. molitor* bioassays, cabbage was used for the dipping assays. A constant concentration of BtA (1.25 mg (=10^8^ spores)/mL) was mixed with XhE (4 mg/mL) for 5 min. Then, the leaf or diet was removed and kept under a clean bench for drying for 10 min and then placed into a Petri dish (90 × 15 mm). After 24 h, the treated leaf or diet was replaced with a fresh one. For the mosquito bioassay, 10 larvae were added to each well of a 12-well Costar culture plate (Corning, Lowell, MA, USA). Then, BtA+XhE was added to each well at a final volume of 1 mL. Dead larvae were counted daily for up to 3 days. Each treatment was replicated three times with 10 larvae for an experimental unit.

### 2.11. Data Analysis

The median lethal concentration (LC_50_) or time (LT_50_) was calculated using the EPA Probit Analysis Program, ver. 1.5 (U.S. Environmental Protection Agency, Washington, DC, USA). All experiments were performed three times. Control efficacy data were analyzed by arcsine transformation (ANOVA). All assay data were analyzed using PROC GLM of the SAS program [38]. All the data were plotted as the mean ± standard error using Sigma Plot (Systat Software, Point Richmond, CA, USA). The TUKEY test was used to compare the means with a Type I error of 0.05.

## 3. Results

### 3.1. Screening for the Optimal Bt Strain to Control S. exigua

The insecticidal activity of four different Bt strains against four different insect species was evaluated (Figure 1). All Bt strains showed specific insecticidal potential to the target insects by exhibiting dose–mortality responses in a dose-dependent manner (Figure 1a). In three Bt strains, 10^8^ spores/mL was the minimum concentration to achieve maximal insecticidal activity, while 10^7^ spores/mL was the least effective concentration in BtT against *P. brassicae*. Then, different Bt strains were applied to *S. exigua* to compare their insecticidal activities at a test concentration of 10^8^ spores/mL (Figure 1b). All four Bt strains exhibited insecticidal activity against *S. exigua* larvae 3 days after treatment, in which BtA was the most potent.

### 3.2. Culture Broth of X. hominickii ANU101 Enhances BtA Toxicity

To enhance BtA insecticidal activity against *S. exigua*, XhE was added to the Bt spores. The mixture of BtA+XhE significantly enhanced the insecticidal activity of BtA (Figure 2a). The mixed treatment also accelerated the speed-to-kill measured by the LT_50_, in which BtA+XhE showed LT_50_ at 44.1 h (95% confidence interval (CI): 33.6~57.8; X^2^ = 0.996; *p* = 0.1991), whereas that of BtA alone was 89.81 h (95% CI: 65.9~122.3; X^2^ = 0.930; *p* = 0.8182). In this assay, XhE alone did not show any significant mortality against *S. exigua*.

To analyze the enhanced effect of XhE on BtA toxicity, four bacterial secondary metabolites of *Xenorhabdus* spp. were added to BtA (Figure 2b). 3-Ethoxy-4-methoxy phenol (EMP) and oxindole (OXI) were identified in Xh culture broth [30]. Benzylideneacetone (BZA) was identified in both *Xenorhabdus* and *Photorhabdus* [39]. GameXPeptide-A (GXP) is also known to be produced by *Xenorhabdus* and *Photorhabdus* [40]. These metabolites significantly (*F* = 16.48; df = 4, 40; *p* < 0.0001) enhanced BtA toxicity in a dose-dependent manner. Although the enhanced BtA toxicity varied between the four metabolites (*F* = 3.78; df = 3, 40; *p* = 0.0178), their enhanced patterns were not much different (*F* = 0.77; df = 12, 40; *p* = 0.6809).

### 3.3. BtA Induces Humoral and Cellular Immune Responses in S. exigua

The effect of BtA on humoral and cellular immune responses was assessed in L5 larvae of *S. exigua* by oral administration (Figure 3a). In the humoral immunity assay, the bacterial challenge using a non-pathogenic *E. coli* strain induced the expression of 10 AMP genes in *S. exigua* larvae that were not infected with BtA (Figure 3b). Interestingly, the *E. coli* challenge to BtA-infected larvae induced four other AMP genes, gloverin, lysozyme, transferrin II, and attacin 2, by 2~8-fold compared to control. In the cellular immunity assay, nodule formation was compared between controls and BtA-infected larvae (Figure 3c). Larvae without BtA treatment exhibited around 58 nodules in response to the *E. coli* challenge. This nodule formation was greatly increased in the larvae infected with BtA.

### 3.4. Suppression of the BtA-Induced Immune Responses by XhE and Four Secondary Metabolites

To explain the enhanced virulence of BtA by the addition of XhE or bacterial metabolites, the suppressive effects of the bacterial metabolites on the immune responses induced by BtA treatment were assessed (Figure 4). As expected, BtA treatment increased the expression level of four AMPs (Figure 4a). However, XhE treatment, along with BtA, significantly suppressed the induction of the four AMP gene expression levels. Similar suppressions were observed in the mixed treatment of BtA and each of the bacterial metabolites. In the cellular immunity assay using nodule formation, the oral administration of XhE or the bacterial metabolites, along with BtA, also suppressed nodule formation in *S. exigua* L5 larvae (Figure 4b).

### 3.5. Field Assay of the BtA+XhE Mixture

The enhancement in BtA virulence by adding XhE was tested in field conditions in a greenhouse cultivating Welsh onions infested with *S. exigua*. To apply the bacterial mixture, Xh culture broth and Bt spores were freeze-dried, respectively, and mixed at different ratios. The optimal mixture ratio was 1:4, which contained 10^8^ spores/mL and 4 mg/mL XhE in a final spray solution in a laboratory test (Figure 5a). This optimal mixture was applied to *S. exigua* larvae infesting Welsh onions in fields (Figure 5b). The BtA+XhE mixture exhibited improved control efficacy compared to BtA treatment alone 3 days after treatment (Appendix A). Seven days after treatment, the mixed treatment gave about 75% control efficacy compared to controls, while BtA alone gave only 45% control efficacy.

### 3.6. Control Spectrum of BtA+XhE

The improved Bt insecticide, made by mixing BtA and XhE, was tested in different insects to determine its control spectrum (Figure 6). Although BtA+XhE produced significant mortality in a wide range of insects, its control efficacy was much more evident in lepidopteran insects than in others. In lepidopteran insects, BtA+XhE was much more effective in controlling *Spodoptera*-associated species, including *S. exigua*, *S. litura*, and *S. frugiperda*.

## 4. Discussion

This study designed a program to develop an effective Bt microbial insecticide based on the immunosuppression of target insects by the addition of the bacterial metabolites of *X. hominickii*. Initially, four different Bt strains were screened to select the most potent Bt strain to control the target insect pest, *S. exigua*.

All four Bt strains potently killed their specific target insects, in which BtA was the most virulent strain against *S. exigua*. This virulence specificity was likely to be a specific interaction between Cry toxin and its binding site(s) in the target insect [41]. When Bt spores are consumed by the target insects, Cry toxins associated with spores are released in the midgut lumen and cleaved by insect digestive enzymes to become active, protease-insensitive toxin proteins. The active toxins migrate to the ectoperitrophic space and interact with Bt-binding site(s) on the membrane of the midgut, in which an aggregation of toxins into oligomers occurs, which disrupts the epithelial membrane by pore formation, resulting in fatal cell lysis called a colloid-osmotic lysis [14,42]. Thus, the specific interaction between Cry toxins and their binding targets on the gut epithelium is essential for Bt pathogenicity. For example, *P. xylostella* larvae are more susceptible to CryIA (a main Cry toxin of BtK) than Cry1C (a main Cry toxin of BtA), whereas *S. exigua* larvae are more susceptible to Cry1C than Cry1A [43,44].

Both cellular and humoral immune responses were significantly induced in *S. exigua* larvae in response to BtA infection. Four AMP genes (gloverin, lysozyme, transferrin II, and attacin II) were highly up-regulated in Bt-infected larvae compared to non-infected larvae in response to the same bacterial challenge. Notably, gloverin is a glycine-rich and heat-stable AMP and showed the potential to defend against Bt virulence in *P. xylostella* [45]. BtA infection enhanced the cellular immune response measured by nodule formation of *S. exigua*. Nodulation was effective in defending against infections with various microbial pathogens including Bt [4,46]. In *S. littoralis*, the 102 Sl gene is essential for nodule formation and plays a crucial role in defending against Bt infection [15]. This suggests that Bt infection stimulates the immune system of the target insects to defend against bacterial infections with humoral and cellular immune responses. Thus, there is a question as to how oral Bt infection induced hemocoelic immune responses. In our assay, the hemocoelic immune assays were performed 24 h after BtA oral administration. A previous study showed that the midgut lesion due to Bt infection was evident after 48 h in *S. exigua*, at which time the intestinal commensals entered the hemocoel [16]. Thus, 24 h after Bt oral infection might be too early for the enteric microbes to enter the hemocoel. This suggests that a signaling molecule(s) to the immune systems in the hemocoel mediated the Bt infection in the gut lumen. A recent study introduced dorsal switch protein 1 (DSP1) as an insect damage-associated molecular pattern (DAMP) [30]. Upon immune challenge by the entomopathogenic bacterium, *Serratia marcescens*, DSP1 was released from the nucleus of gut epithelium to the hemocoel and activated PLA_2_ to synthesize eicosanoids in thrips [47]. Eicosanoids then mediated both cellular and humoral immune responses [12]. It was interesting to observe the inhibitory activity of EMP against the immune defenses induced by BtA infection because EMP is known to bind to DSP1 and prevent its release from the nucleus upon immune challenge [48]. This suggests a role of DSP1 in mediating immune responses upon BtA infection in *S. exigua*. Thus, the oral administration of BtA would infect the gut epithelium, which induces the release of DSP1 from the damaged cells to the hemocoel. The released DSP1 in the hemocoel would then mediate Bt infection signals by activating both cellular and humoral immune responses via eicosanoids.

The immune responses induced by Bt infection were effectively suppressed by the bacterial metabolites of *X. hominickii*. The culture broth (=XhE), presumably containing the secondary metabolites, suppressed the cellular and humoral immune responses induced by BtA infection. Four bacterial metabolites also inhibited the immune responses, in which two compounds (oxindole and EMP) were confirmed as bacterial metabolites of *X. hominickii* and the other two compounds (BZA and GXP) are regarded to be present in the bacterial culture broth because of their wide occurrence in *Xenorhabdus* and *Photorhabdus* [39,49]. BZA was the first compound to show the specific inhibition of PLA_2_ from *X. nematophila* [50]. Subsequent studies identified six other PLA_2_-inhibitory compounds [39] in different *Photorhabdus* and *Xenorhabdus* strains. These compounds were produced during different growth phases and inhibited different molecular targets associated with immune responses. For example, BZA more efficiently inhibited hemocytic nodule formation, whereas oxindole was more effective at inhibiting phenoloxidase [4]. GXP is a cyclic pentapeptide and is widespread in *Photorhabdus* and *Xenorhabdus* [40]. Its chemical similarity with sansalvamide-A, which inhibits heat shock protein 90 and induces cytotoxicity in mammals [51], suggests its immunosuppressive activity in insects. Indeed, GXP inhibited PLA_2_ and the cellular and humoral immune responses of *S. exigua* [2]. The immunosuppressive activities of the bacterial metabolites support our hypothesis that any suppression of immune responses would lead to enhanced Bt virulence because the addition of XhE significantly enhanced BtA virulence against *S. exigua*.

The enhancement of BtA virulence by the addition of XhE was confirmed in the field application. In this field assay, we used 3 mg/mL of XhE along with 10^8^ spores/mL of BtA. Less XhE did not show higher control efficacy in the laboratory assay, suggesting an effective amount of the bacterial metabolites is necessary to inhibit the immune responses of *S. exigua*. The laboratory and field assays suggested that the mixture of BtA+XhE was an IBC agent, as explained in the Introduction. This IBC was also effective against other *Spodoptera*-associated insect pests such as *S. litura* and *S. frugiperda*. However, it did not effectively control other insects. This differential toxicity of BtA+XhE might be due to a specific interaction between Bt Cry toxins and their binding site(s) on the gut epithelium of the target insects [41]. Thus, replacing BtA in the IBC agent with other Bt strains might control other insect species. For example, a BtT+XhE or BtI+XhE mixture might control coleopteran or dipteran insect pests. This tailor-made approach needs to be further explored. Altogether, this study demonstrated the enhancement of *B. thuringiensis* virulence by suppressing the induction of cellular and humoral immune responses in the lepidopteran insect, *S. exigua*.

## Figures and Tables

**Figure 1 insects-13-00860-f001:**
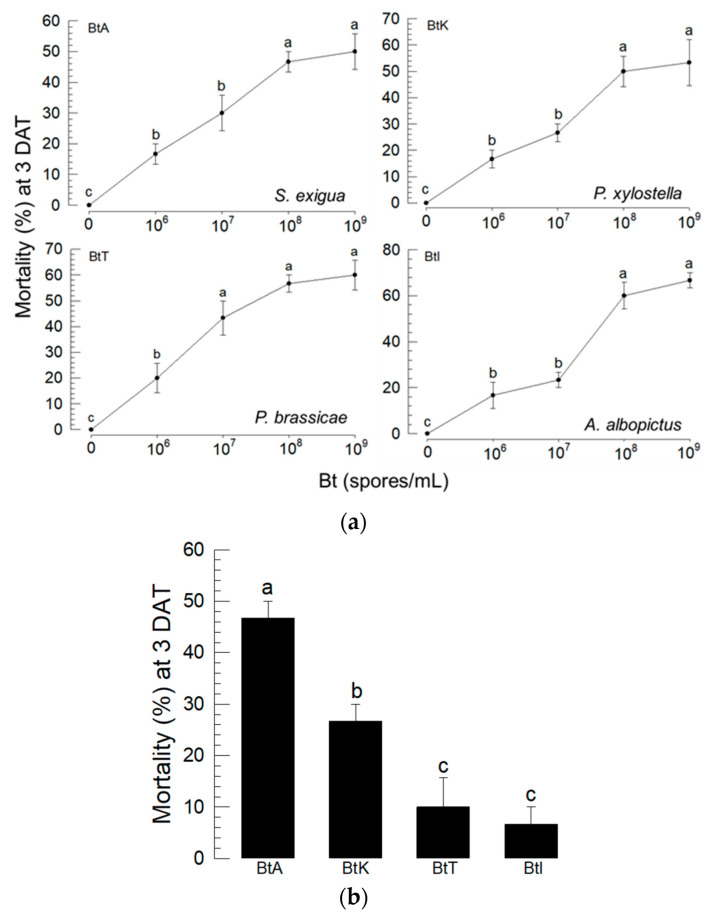
Relatively high virulence of *B. thuringiensis aizawai* (BtA) against *S. exigua*. (**a**) Virulence of four Bt strains of BtA, *B. thuringiensis kurstaki* (BtK), *B. thuringiensis tenebrionis* (BtT), and *B. thuringiensis israelensis* (BtI) against *S. exigua*, *P. xylostella*, *P. brassicae*, and *A. albopictus* larvae, respectively. (**b**) Comparative analysis of four Bt strains (10^8^ spores/mL) against *S. exigua*. An experimental unit consisted of 10 larvae. Each dose was replicated three times. Mortality was assessed 3 days after treatment (DAT). Different letters above the standard error bars indicate significant differences between the means at Type I error = 0.05 (TUKEY test).

**Figure 2 insects-13-00860-f002:**
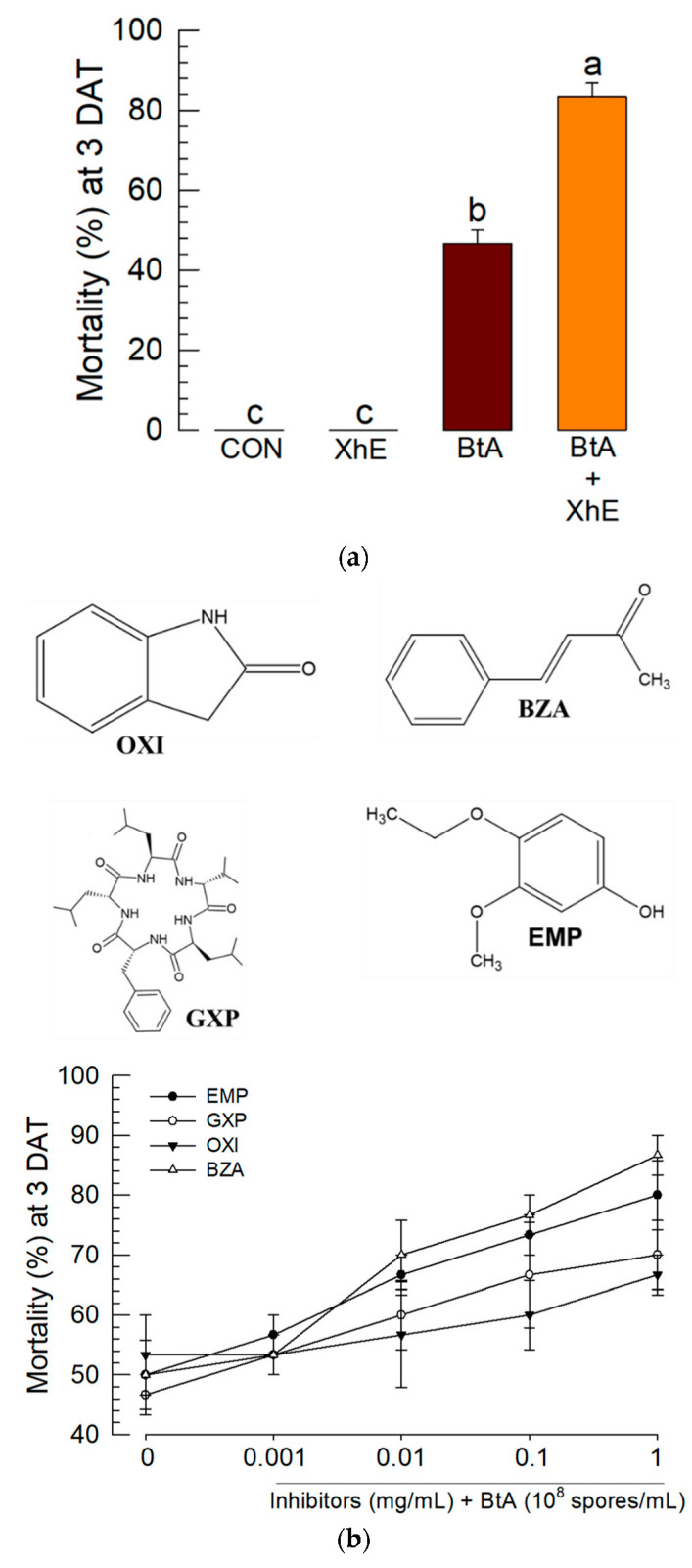
Enhanced virulence of *B. thuringiensis aizawai* (BtA) against *S. exigua* by the addition of the bacterial metabolites of *X. hominickii*. (**a**) Effect of the bacterial culture broth of *X. hominickii* (XhE) on BtA virulence. *S. exigua* L3 larvae were treated with BtA (10^8^ cells/mL) or its mixture (BtA+XhE) with XhE (4 mg/mL) by the leaf-dipping method. (**b**) Effect of the bacterial metabolites on BtA virulence. *S. exigua* L3 larvae were treated with BtA (10^8^ cells/mL) or its mixture with different doses of the four bacterial metabolites (1 mg/mL) by the leaf-dipping method. An experimental unit consisted of 10 larvae. Each treatment or dose was replicated three times. Mortality was assessed 3 days after treatment (DAT). Different letters above the standard error bars indicate significant differences between the means at Type I error = 0.05 (TUKEY test).

**Figure 3 insects-13-00860-f003:**
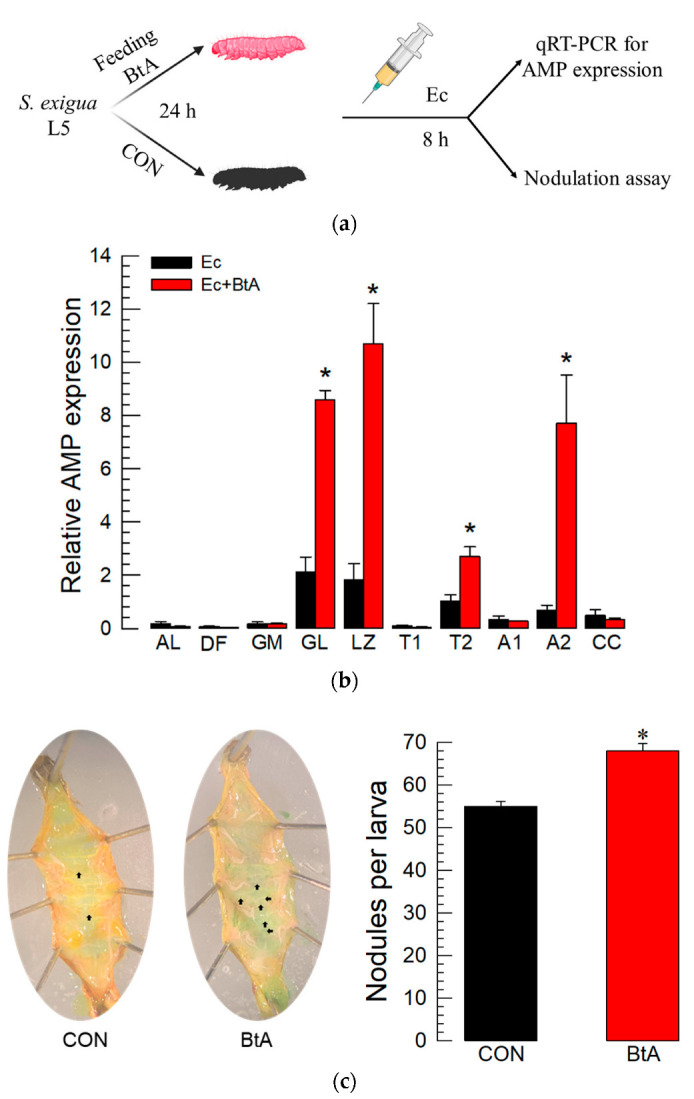
Induction of immune responses to *B. thuringiensis* in *S. exigua*. (**a**) A schematic diagram of the experimental design. L5 larvae were fed *B. thuringiensis aizawai* (BtA, 10^8^ spores/mL) by the leaf-dipping method. For the controls (CON), L5 larvae were fed the diet without BtA treatment. After 24 h, the larvae were injected with *E. coli* (Ec, 1.75 × 10^5^ cells/larva) and incubated for another 8 h at 25 °C. Subsequently, humoral immunity was assessed by quantifying the expression levels of 10 antimicrobial peptide (AMP) genes and cellular immunity was assessed by nodule formation. (**b**) The humoral immunity assay showed the expression of 10 AMP genes: apolipophorin III (AL), defensin (DF), gallerimycin (GM), gloverin (GL), lysozyme (LZ), transferrin I (T1), transferrin II (T2), attacin 1 (A1), attacin 2 (A2), and cecropin (CC). Fat body tissue was used for RNA extraction. Each treatment was replicated three times with independent insect samples. (**c**) Cellular immunity assay using nodulation. Arrows in the photos indicate nodules attached to the gut or fat body in the hemocoel. Three insects were tested in each treatment. An asterisk above the standard error bars indicates a significant difference between the treatment and controls at Type I error = 0.05 (TUKEY test).

**Figure 4 insects-13-00860-f004:**
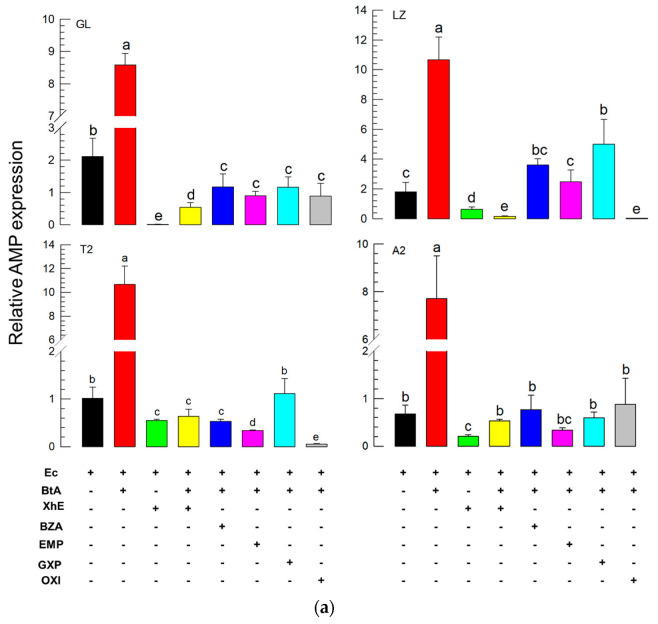
Immunosuppressive effects of the bacterial metabolites of *X. hominickii* on the cellular and humoral immune responses induced by *B. thuringiensis aizawai* (BtA) infection in *S. exigua* L5 larvae. Experiments followed the method described in Figure 3a, in which larvae were treated with bacterial culture broth (XhE, 4 mg/mL) or four bacterial metabolites (BZA, EMP, GXP, and OXI, 1 mg/mL), along with BtA (10^8^ cells/mL). The subsequent immune challenge used an injection of *E. coli* (Ec, 1.75 × 10^5^ cells/larva) and incubation for another 8 h at 25 °C. (**a**) Suppression of the humoral immune response induced by the addition of XhE or four metabolites to BtA treatment. This analysis used four antimicrobial peptide (AMP) genes highly induced upon BtA infection: gloverin (GL), lysozyme (LZ), transferrin II (T2), and attacin 2 (A2). Fat body tissue was used for RNA extraction. Each treatment was replicated three times with independent insect samples. ‘+’ and ‘-’ represent presence and absence of a test sample, respectively. (**b**) Suppression of the cellular immune response induced by the addition (‘+’) of XhE or four metabolites to BtA treatment. This analysis used nodule formation. In each treatment, three insects were tested. Different letters above the standard error bars indicate significant differences between the means at Type I error = 0.05 (TUKEY test). Different color bars indicate different treatments.

**Figure 5 insects-13-00860-f005:**
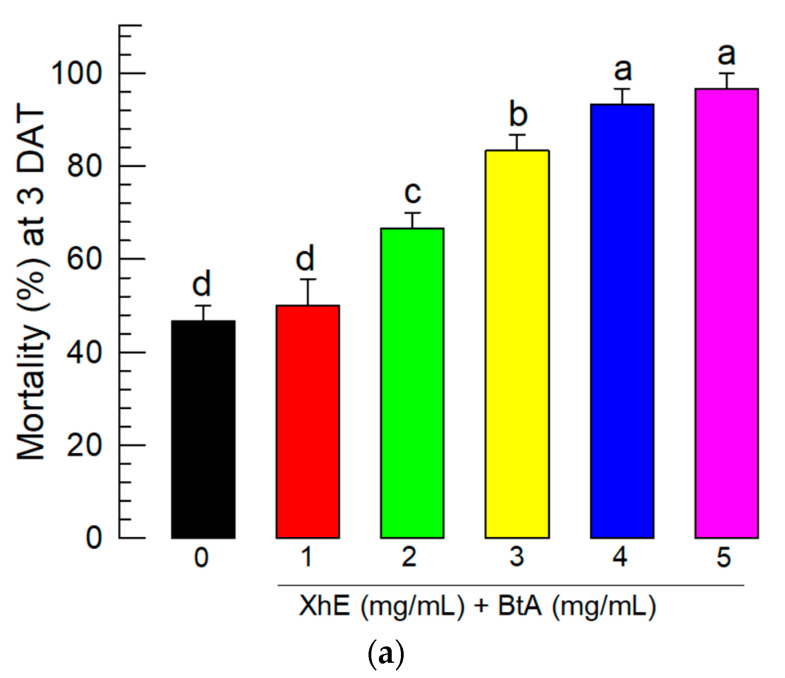
Control efficacy of the BtA+XhE formulation against *S. exigua*. Bacterial cells of *B. thuringiensis aizawai* (BtA) and the bacterial culture broth of *X. hominickii* (XhE) were freeze-dried and mixed in a mass ratio for the application formulation. (**a**) Laboratory test to determine the optimal mixture ratio of BtA and XhE. A one-mL volume of spraying suspension contained 1.25 mg of BtA and 10^8^ spores along with varying amounts of XhE. Each experimental unit consisted of 10 L3 larvae and was replicated three times. Mortality data were assessed 3 days after treatment (DAT). (**b**) Control efficacy of BtA+XhE against *S. exigua* infesting Welsh onions in a greenhouse. The spraying suspensions were XhE (4 mg/mL), BtA (1.25 mg (=10^8^ spores)/mL), or their mixture (BtA+XhE). The initial insect density was 26~31 larvae per experimental unit (Appendix A). Each treatment was replicated three times. Control efficacy was measured at 7 DAT. Different letters above the standard error bars indicate significant differences between the means at Type I error = 0.05 (TUKEY test). Different color bars indicate different treatments.

**Figure 6 insects-13-00860-f006:**
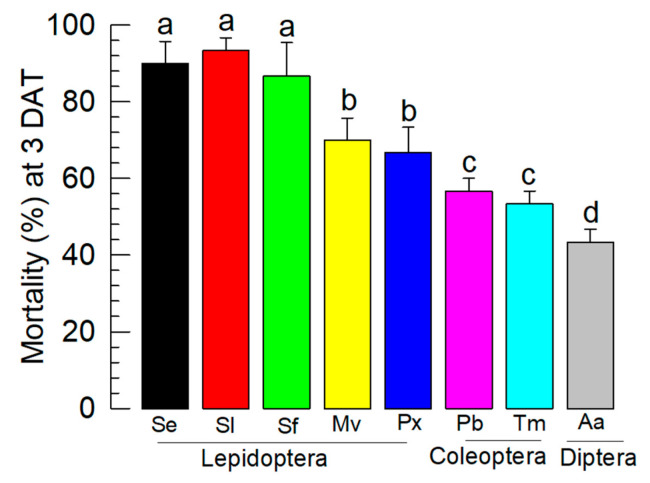
Control spectrum of the BtA+XhE formulation against *S. exigua*. Bacterial cells of *B. thuringiensis aizawai* (BtA) and the bacterial culture broth of *X. hominickii* (XhE) were freeze-dried and mixed at a 1:4 mass ratio for application. All test insects used L3 larval stage for the feeding assay. Se, *S. exigua*; Sl, *S. litura*; Sf, *S. frugiperda*; Mv, *M. vitrata*; Px, *P. xylostella*; Pb, *P. brassicae*; Tm, *T. molitor*, and Aa, *A. albopictus*. Mortality data were assessed 3 days after treatment (DAT). Different letters above the standard error bars indicate significant differences between the means at Type I error = 0.05 (TUKEY test). Different color bars indicate different treatments.

## Data Availability

All data are contained within the article.

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
