# Peer review of "Integrated Biological Control Using a Mixture of Two Entomopathogenic Bacteria, Bacillus thuringiensis and Xenorhabdus hominickii, against Spodoptera exigua and Other Congeners"

_insects, 2022, doi:10.3390/insects13100860_

Round 1
Reviewer 1 Report
2. Materials and Methods
2.1. Insect Rearing
Larvae were grown to the L3 instar stage for the bioassays.
Question: Why did you choose L3 instar larvae for bioassays? Why not choose L2, L4 or L5 instar larvae? But in 2.4, 2.5, 2.6 part, you choose L5 larvae, not L3 instar.
All insects were reared at 25 ± 2oC and relative humidity of 60 ± 5%.
Here the Light : dark cycle used for rearing insect should be provided.
2.6. Antimicrobial Peptide (AMP) Gene Expression Analysis
To assess AMP gene expression, S. exigua L5 larvae were treated with heat-killed E.coli as described above.
Here, E.coli : should use italics for the species names
3. Results
3.3. BtA Induces Humoral and Cellular Immune Responses in S. exigua
Larvae without BtA treatment exhibited around 58 nodules in response to the E. coli challenge. This nodule formation was greatly increased in the larvae infected with BtA.
Question: Here, as shown in Fig 3C, it seemed that the nodule formation was not increased “greatly “?
4. Discussion
Initially, four different Bt strains were screened to select the most potent Bt strain to control the target insect pest, S. exigua.
Question: why did you choose these four kinds of the Bt strains to screen, not other strains? What is the reason?
This study demonstrated the enhancement of B. thuringiensis virulence by suppressing the induction of cellular and humoral immune responses in the lepidopteran insect, S. exigua.
As we known, B. thuringiensis is used widely for control of lepidopteran insect, H. armigera. Why don't you choose H. armigera to test your hypothesis?
Author Response
Comment #1-1: Why did you choose L3 instar larvae for bioassays? Why not choose L2, L4 or L5 instar larvae? But in 2.4, 2.5, 2.6 part, you choose L5 larvae, not L3 instar.
Response: Yes, we used different stages for insecticide bioassays and for immunological assays. This information is added as follows: “Larvae at L3 stage were used for the bioassays while L5 larvae were used for immunological assays.”
Comment #1-2: All insects were reared at 25 ± 2oC and relative humidity of 60 ± 5%. Here the Light : dark cycle used for rearing insect should be provided.
Response: Corrected as follows: “All insects were reared at 25 ± 2oC, relative humidity of 60 ± 5%, and 16:8 (L:D) photo-period.”
Comment #1-3: 2.6. Antimicrobial Peptide (AMP) Gene Expression Analysis
To assess AMP gene expression, S. exigua L5 larvae were treated with heat-killed E.coli as described above. Here, E.coli : should use italics for the species names
Response: Corrected as suggested
Comment #1-4: 3.3. BtA Induces Humoral and Cellular Immune Responses in S. exigua
Larvae without BtA treatment exhibited around 58 nodules in response to the E. coli challenge. This nodule formation was greatly increased in the larvae infected with BtA.
Here, as shown in Fig 3C, it seemed that the nodule formation was not increased “greatly “?
Response: Corrected as follows: “This nodule formation was much increased in the larvae infected with BtA.”
Comment #1-5: Initially, four different Bt strains were screened to select the most potent Bt strain to control the target insect pest, S. exigua.
why did you choose these four kinds of the Bt strains to screen, not other strains? What is the reason?
Response: Four Bt strains have different Cry toxins. From our assay showed that BtA was the most potent to S. exigua because it specifically produces Cry1C compared to other Bt strains. This is explained in Discussion.
Comment #1-6: This study demonstrated the enhancement of B. thuringiensis virulence by suppressing the induction of cellular and humoral immune responses in the lepidopteran insect, S. exigua. As we known, B. thuringiensis is used widely for control of lepidopteran insect, H. armigera. Why don't you choose H. armigera to test your hypothesis?
Response: As mentioned before, Cry1C is known to be highly potent to Spodoptera. To prove this, we used different spodopteran species.

Reviewer 2 Report
Comments on “Integrated Biological Control Using a Mixture of Two Entomopathogenic Bacteria…” by Hrithik et al.
The manuscript submitted by Hrithik et al. deal with a topic that has a great potential applied interest, that is, an enhancement of integrated biological control by combining the action of two entomopathogenic bacteria. The Authors offer some interesting data and information that add to a previous solid ground of research on the same subject. At the same time, I believe that some aspects of data analysis could and should be improved.
My main concern is about some details of statistical analysis. My understanding of what stated in paragraph 2.11 (Data Analysis) is that assay data were submitted to ANOVA trough PROC GLM of the SAS program, using arcsine transformation when percentages were involved, while LSD was used as a mean comparison test.
First, I noticed that the F value of the ANOVA was not always presented (like in Results 3.1 Screening for the Optimal Bt Strain…, or the F values relative to the data presented in figures 4, 5 and 6). F values should always be presented and the problem is accentuated by the choice of LSD which is too liberal as a mean comparison test. When the LSD comparison is between two means only (as in data presented in Fig.3) this is not a problem because in this case the comparison-wise Type I error rate equals the experiment-wise Type I error rate. But if you use LSD with a higher number of compared means the experiment-wise error rate increases and will exceed the alfa level of 0.05.
When the comparison is among 4 mean values (as for the data showed in Fig. 1) the experiment-wise error rate will rise to a value of about 0.19, with 6 mean values (as in Fig. 5a) the experiment-wise error rate will be about 0.26, and with 8 compared means (as in Fig. 4 and 6) it will rise to about 0.34. In other terms, LSD is too liberal a test to be used when comparing many means and the possibility of declaring as different two means that statistically are not different is too high. I would recommend using either SNK or Tukey tests instead of LSD. In any case, F values should aways been presented. I do not think that changing the mean comparison test would change the overall picture, but I believe that statements about significance in means difference will be more accurate.
Other minor remarks and suggestions:
- The first time a species name is presented in the MS the genus name should not be abbreviated (as in row 75, it should be Xenorhabdus nematophila; in row 92 Spodoptera litura and in row 95 Xenorhabdus hominickii)
- In row 222 correct “arcsine” instead of “arsine”
- The position of the sentence in rows 275-277 “S. exigua L3 larvae were treated … by the leaf-dipping method” could confuse the reader because it seems to be relative to the data presented in Fig 2b (the previous sentence) while it looks more appropriate (treated with BtA OR its mixture) with reference to data of Fig 1A.
Author Response
Comment #2-1: First, I noticed that the F value of the ANOVA was not always presented (like in Results 3.1 Screening for the Optimal Bt Strain…, or the F values relative to the data presented in figures 4, 5 and 6). F values should always be presented and the problem is accentuated by the choice of LSD which is too liberal as a mean comparison test. When the LSD comparison is between two means only (as in data presented in Fig.3) this is not a problem because in this case the comparison-wise Type I error rate equals the experiment-wise Type I error rate. But if you use LSD with a higher number of compared means the experiment-wise error rate increases and will exceed the alfa level of 0.05. When the comparison is among 4 mean values (as for the data showed in Fig. 1) the experiment-wise error rate will rise to a value of about 0.19, with 6 mean values (as in Fig. 5a) the experiment-wise error rate will be about 0.26, and with 8 compared means (as in Fig. 4 and 6) it will rise to about 0.34. In other terms, LSD is too liberal a test to be used when comparing many means and the possibility of declaring as different two means that statistically are not different is too high. I would recommend using either SNK or Tukey tests instead of LSD. In any case, F values should aways been presented. I do not think that changing the mean comparison test would change the overall picture, but I believe that statements about significance in means difference will be more accurate.
Response:
(1) All LSD tests among means were reassessed by TUKEY tests.
(2) F values are added in all appropriate places.
Comment #2-2: The first time a species name is presented in the MS the genus name should not be abbreviated (as in row 75, it should be Xenorhabdus nematophila; in row 92 Spodoptera litura and in row 95 Xenorhabdus hominickii)
Response: Corrected as suggested
Comment #2-3: In row 222 correct “arcsine” instead of “arsine”
Response: Corrected as suggested
Comment #2-4: The position of the sentence in rows 275-277 “S. exigua L3 larvae were treated … by the leaf-dipping method” could confuse the reader because it seems to be relative to the data presented in Fig 2b (the previous sentence) while it looks more appropriate (treated with BtA OR its mixture) with reference to data of Fig 1A.
Response: Yes,I believe it is in an appropriate position because it explains the experiment in Fig. 2.
